# Social Marketing in Promoting Sustainable Healthy Lifestyle among Student Population

**Nikola Milicevic** , **Ines Djokic, Nenad Djokic \*** and **Aleksandar Grubor**

Department of Trade, Marketing and Logistics, Faculty of Economics in Subotica, University of Novi Sad, 24000 Subotica, Serbia; nikola.milicevic@ef.uns.ac.rs (N.M.); ines.djokic@ef.uns.ac.rs (I.D.); aleksandar.grubor@ef.uns.ac.rs (A.G.)

**\*** Correspondence: nenad.djokic@ef.uns.ac.rs; Tel.: +381-21-485-2926

**Abstract:** Although there are some differences in healthy lifestyle measurement, physical activity is an indispensable factor within that construct. By increasing the level of physical activity of the population, the contribution to social sustainability is provided. Social marketing can be considered as a manner to promote behavior change (including increase in physical activity level). It uses commercial marketing tools in delivering social goods. In that context can be explained previous uses of the theory of planned behavior (TPB) in social marketing researches aiming at increasing the level of physical activity of the population. In this paper, the modified TPB model, extended with self-identity and motivation variable, is implemented within the student population of the University of Novi Sad, Serbia, in 2019. The results show that intention to do regular physical activity in the week after the research was directly influenced by behavioral and normative beliefs and self-identity. It was influenced indirectly by students' motivation. The motivation, however, directly affects students' behavioral, normative and control beliefs. Nevertheless, the results differ among genders; although positive at both genders, the effects of normative beliefs and motivation on intention were significant only in female students (0.123 and 0.243, respectively). The authors also provide social marketing implications, i.e., potential activities within social marketing that could be performed in order to encourage students to be more physically active. In addition to belonging to relatively scarce similar researches in domestic context, the wider contribution of this paper can be identified from a methodological aspect, treating the behavioral, normative and control beliefs as formative constructs.

**Keywords:** social sustainability; social marketing; sustainable healthy lifestyle; theory of planned behavior; physical activity; student population; Republic of Serbia



## 1. Introduction

Physical inactivity is one of the greatest global problems for public health, reaching the extent of a pandemic [1]. Hereby, according to the results of Eurobarometer research from 2013, 42% of inhabitants of the European Union never go in for sports, and 17% less than once a week [2]. In the Republic of Serbia, according to the research of the Ministry of Health, only every ninth citizen (11%) spent at least 90 min a week doing sports and recreation during their leisure time [2]. Physical inactivity is not only one of the key causes of numerous illnesses (such as obesity, diabetes, cardiovascular and malignant diseases etc.) but can also influence increase in healthcare cost, unemployment, absenteeism and retirement due to disability [2]. On the other hand, regular physical activity, as an important segment of a healthy lifestyle, has positive impact on mental and physical health, decreasing a risk of depression and all-cause mortality [3].

When it comes to a healthy lifestyle, there are some differences regarding its operationalization [4–11]. However, all cited researches consider physical activity as a part of a healthy lifestyle. In addition, by contributing to increasing physical and psychological

human wellbeing through improving health, physical activity supports achieving social sustainability [12].

Contrary to some expectations, several studies confirmed that the level of physical activity is significantly lower in adulthood compared to the adolescence period. The reason for this may lie in the fact that after finishing high school and starting to work or study, engaging in physical activity is mostly on a voluntary basis [3]. Although the life during studies at the university brings certain freedoms, many students are confronted with academic pressure and lack of time for physical activity. Hence, according to the meta-analysis of a number of researches, physical inactivity among college students ranges from 40 to 50%, representing a serious threat to the health of young adulthood [3].

However, changing physical activity behavior is not easy at all, especially bearing in mind the notion of several researchers according to which individuals would not do that simply at the request of others [13]. A change in behavior and its encouragement is in the focus of social marketing. There are several forms of evidence that point to the role that marketing theory can have in availing the delivery of social change, which, among the others, refers to physical activity as well [14]. Hereby, in the context of social marketing, before taking some actions related to promoting and encouraging physical activity among students, the analysis of their perceptions and beliefs needs to be done.

Having all previously listed in mind, the attention in this paper was dedicated to the relation of physical activity and healthy lifestyle, as well as the use of social marketing for increasing physical activity of the student population. Within the literature review, there is firstly an explanation of sustainable healthy lifestyle and the role of physical activity within it. It is followed by explanations of social marketing and description of its activities, especially its usage for promoting physical activity. In the primary research, there is an examination of physical activity behavioral intention within the student population from Serbia (in the context of the application of the theory of planned behavior (TPB)). Besides the relations between behavioral, normative and control beliefs, on one side, and physical activity intention, on the other, the research model included two more constructs: motivation and self-identity. Hereby, to the knowledge of the authors, behavioral, normative and control beliefs were, for the first time, conceptualized as formative constructs. That conceptualization is in accordance to contemporary scientific recommendations and applying different approach could lead to wrong conclusions. After results, a discussion and conclusion section was presented, including implications related to promoting and encouraging physical activity among students.

## 2. Literature Review and Hypotheses Development

### 2.1. Sustainable Healthy Lifestyle and Physical Activity

It has already been stated that there are some differences in operationalizing a healthy lifestyle construct. In previous researches, it was (positively or negatively) associated with the following elements (or their levels): health checkups [4], physical activity [4–11], smoking [4–11], drinking [4,6,8–11], weight-body mass index [5,6,8–11], diet [5–11], television exposure [8], afternoon nap [8], meeting up with friends [8], number of working hours [8]. In addition, even regarding the consideration of a single element, there can be found differences in cited researches. For example, when it comes to diet, it was estimated, among others, through fruit and vegetable consumption [5], adherence to the Mediterranean dietary pattern [6,8,9] or healthy eating index [7].

Because of the existence of differences in measuring a healthy lifestyle, an attempt was made to create composite "Healthy Lifestyle" measure [15]. Hereby, into account were taken leisure time exercise, eating fruits and vegetables five or more times during a day, sleeping more or equal to 7 h during a 24 h period, not smoking, and not drinking excessively. When it comes to leisure time exercise, respondents were questioned whether in the month prior to research they had, other than their regular job, participated in some of the physical activities or exercises such as running, calisthenics, golf, gardening, or walking for exercise. As for not smoking, respondents who had not smoked 100 cigarettes and

did not in the time of the research smoke every day or some days were considered as nonsmokers. In regards to not drinking excessively, it can be stated that excessive drinking was a combination of two measures—heavy drinking (more than two drinks per day on average for men or more than one drink per day on average for women) and binge drinking (five or more drinks during a single occasion for men or four or more drinks during a single occasion for women) in the 30 days period. For the purpose of generating the composite measure for all five components, the number of desirable behaviors was counted and the total ranged from 0 to 5.

In addition, the components of a healthy lifestyle (with emphasis on adolescence) were identified in another manner—based on review of previous researches and by using qualitative approach and content analysis [16]. Hereby, there were four main categories and twelve subcategories: physical health (mobility, nutrition, proper sleep pattern, reduction of high-risk behaviors, health responsibility, accident prevention and self-care), mental health (stress management, self-fulfillment, positive thinking and mindfulness), social health (interpersonal relationships) and spiritual health (spiritual growth).

When it comes to students' healthy lifestyle, great attention is devoted to physical activity [17]. Therefore, an educational technology of managing students' healthy lifestyle is proposed. It consists of three stages with following objectives (respectively): to acquire knowledge and broaden understanding of healthy living, to develop healthy lifestyle habits in the course of studies, and to involve remedial health care activities.

The connection between physical activity and social sustainability is explained in the literature as well [12]. Cited authors start from definition of social sustainability as "a positive condition within communities, and a process within communities that can achieve that condition" [18] (p. 23) i.e., from understanding that "social sustainability means meeting the needs for human well-being" [19] (p. 63). In addition, they rely on consideration of wellbeing in the context of fulfilling human needs [20] consisting of physical, emotional and social elements including, among others, exercise [19] that can, through positive effects on health, improve physical and psychological wellbeing [21].

It can be concluded that healthy lifestyle was in different researches brought into connection with numerous positive effects: reducing the risk of developing pancreatic [6] or breast cancer [7], lowering the risk of developing primary cardiovascular disease [8], and concretely peripheral artery disease [9] and stroke [10], reducing premature mortality and prolonging life expectancy [11], better health, lower rates of chronic disease and better access to health care [15].

### 2.2. Social Marketing

A need to recognize the authenticity and legitimacy of social marketing as a separate discipline has existed for years [22]. It appeared in practice as early as the 1960s with the beginnings of the family planning program, whereas in marketing literature, it has been present for several decades already [23].

Kotler and Zaltman were among the first authors to address the issue of social marketing. They pointed to the development and significance of this discipline in their 1971 paper [24]. In this period, among others, marketers helped the work of humanitarian organizations, advised churches on how to increase congregations, charities on how to raise greater funds, and museums and symphonies on how to attract interest of a higher number of sponsors. However, according to those authors, social marketing is much more than promotional activities, that is, than communication mix. They define it as "design, implementation, and control of programs calculated to influence the acceptability of social ideas and involving considerations of product planning, pricing, communication, distribution, and marketing research" [24] (p. 5).

That social marketing has become a universally accepted discipline as confirmed by numerous examples, both in a conceptual–theoretical sense (several books and chapters were published, a journal has been started, centers have been established, training programs and institutes have been formed) and in a practical sense (approaches regarding social

marketing have been accepted by a large number of national, international, consultancy and marketing organizations) [23].

Moreover, new views of the given issue have appeared. Social marketing is not viewed as a set of techniques but rather as a process of development of programs of social changes modeled analogously to the marketing processes in the private sector [23]. The focus is on changes in behavior, needs of clients and creation of exchanges encouraging that behavior. In accordance with that, there are several criteria based on which a certain approach can be classified as social marketing [23]:

- A change in behavior is the basic criterion when designing and evaluating activities (interventions);
- Carrying out research in order to understand the needs of the target audience, test elements before their application and implement control of the development of activities;
- Segmenting target audience for more efficient and effective use of resources;
- Creating attractive exchanges with target audience;
- Application of marketing mix elements ("4P")—offer of attractive useful packages (products), with cost (price) cutting wherever possible, simplifying and facilitating exchange (place), combined with sending message through media adapted to target audience (promotion);
- Special attention is devoted to competition related to desired behavior.

Having as a base the definition of the above cited author, according to which social marketing represents application of commercial marketing techniques in analyzing, planning, execution and evaluation of programs, designed to influence voluntary behavior of target audience in order to improve the position (welfare) of individuals and the society, certain modifications were proposed in order to add the adjective "involuntary" to "voluntary behavior" and to include decision makers who influence welfare into the definition [25]. Thus, for instance, a marketing campaign resulting in establishing regulations on production and consumption of less saturated fat product would influence legislators' voluntary behavior and involuntary behavior of producers and their buyers [25].

Social marketing refers to the application of marketing tools in resolving health, social and other problems, resulting in positive social change [22]. Similar to commercial conditions, the final goal is change in behavior. However, while success in profit-making organizations is measured through achieved sale, brand recognition or market share, the basic criteria for social marketing are achieving individual and social welfare [22].

Social marketing can also be related to achieving and promoting general welfare [26]. Its strategy must not be subdued to the influence of conventional marketing. In addition, social marketing strategies ought to exceed promoting better and healthier choices for individuals and encourage critical thinking, political engagement and social action [26]. A more intensive inclusion of people into political, social and other questions of the community is a prerequisite for initiating more significant, positive changes in society.

That social marketing should rise above changes in individual behavior is also pointed out [27]. Approach based on segmentation and application of marketing mix primarily deals with the visible symptoms of health and other problems in the environment, influencing the changes in behavior of a certain target audience. However, the influence of a given social marketing approach is narrow; its reach is limited, and the effects are insufficient to achieve a sustainable social change in the case of complex and large problems. Consequently, the need arises to establish a systemic approach to studying this topic (including scales, causality principle, etc.) with the aim to implement a large number of actions (interventions), both for society as a whole and for its different segments [27].

Social marketing can be viewed from the aspect of various principles [28]. They include a clearly defined problem, orientation to citizens, focus on behavior, theoretical basis, exchange of values, integrated activities, research and collaboration. Despite the fact that most of these principles represent a part of best practices approach, in business operations and pubic healthcare, the following three can be singled out [28]:

- Orientation to citizens—insight into citizens' daily life can provide a view of effects of certain policies and programs, and also their potential changes in behavior;
- Focus on behavior—encourages positive behavior and changes in the function of social welfare;
- Exchange of values—to achieve the desired behavior, it is necessary to offer a certain value and exchange it with target audience.

One of the notions appearing frequently in the definition of social marketing is "social good" [29]. Its interpretation depends to a great extent on ethnical and political factors of the environment, and also on the development of social institutions. Presenting social good as one of the key outcomes of social marketing can arise from relating the given notion to the Universal Declaration of Human Rights, passed in 1948 by the United Nations [29]. It consists of 30 articles, regulating the basic human rights. Using the Declaration to establish whether a given behavior or social marketing program is in compliance with the social goods implies respecting all rights, which is not a very simple task. Bearing in mind that one campaign aimed at achieving "social good" may jeopardize freedoms or dignity of some groups (for instance, in the case of anti-smoking campaigns), it is necessary to strive to establish a compromise in accordance with the valid ethical and political principles [29].

### 2.3. Social Marketing and Physical Activity

When developing social marketing programs in accordance with the needs of various groups, segmentation theory can be used to achieve social change as effectively as possible [30]. Previously cited research has shown that, from the aspect of doing physical activities, three basic segments can be differentiated: women positivists, active men and the young and motivated. The changes in the level of exercise in a period of a year pointed to positive movements in all three segments in relation to the number of exercise sections, recreation or sport and time spent for listed activities [30].

When designing social marketing campaigns, various formative research may be of assistance. In a formative study, the perceived benefits and barriers related to physical activity were analyzed, based on which respondents were classified in four segments (high benefits/high barriers, low benefits/low barriers, low benefits/high barriers and high benefits/low barriers) [31]. People with small perceived barriers reported more physical activities in comparison with those with greater perceived barriers to exercise. Moreover, people with high perceived barriers and small perceived benefits related to physical activity are characterized by the lowest level of health-consciousness, as well as a high level of body mass index (BMI).

Various cognitive theories may have a significant application within social marketing. In search of an adequate theoretical model of exercise behavior and physical activity, three theories were tested and compared: theory of reasoned action (TRA), theory of planned behavior (TPB) and modified TPB (with an additional link from subjective norms to attitudes) [32]. All models had satisfactory indicators' fit, while the standard TPB model was proved to be superior. In accordance with this, their results showed that all three constructs (attitude, subjective norm and perceived behavioral control) make a positive impact on exercise intention with a particularly expressed effect of attitude.

The possibility of application of the theory of planned behavior for developing strategies of promoting physical activities is also emphasized [33]. In addition to attitudes, subjective norms and perceived behavioral control as direct predictors of intention, cited authors also included beliefs related to goals and barriers into the analysis. This paper looks into the effects of the above mentioned direct and indirect predictors of physical activity intention on the total sample and by individual demographic groups. Certain differences were identified among them, which can be used in defining promotion strategies of encouraging physical activity for different age-gender segments.

In a certain number of papers, TPB was applied in the analysis of intentions and behaviors related to physical activity of younger populations, especially bearing in mind the increasingly present problem of children obesity. Thus, through the implementation of an

extended version of the TPB approach, the given topic was researched on elementary school fifth-graders [34]. While doing this, beliefs were used within the TPB theory instead of three constructs of intention (attitude, subjective norms and perceived control of behavior). The analysis also included other variables that may influence intentions or behavior of children, including descriptive norm, self-identity and facilitating factors. Moreover, the research included indicators related to parental support, level of their physical activity, body mass index (BMI) and sedentary activities. The results showed that children's physical activity correlates positively to intention and self-identity, whereas the basic determinants of physical activity intentions are self-efficiency, self-identity, positive behavioral beliefs, and gender. Additional analysis of the intentions and key beliefs from the aspect of gender was performed, pointing to certain discrepancies between boys and girls.

In researching adolescents' intentions and behavior related to physical activity, TPB model was extended with self-identity and motivation [35]. In addition to confirming the planned behavior in terms of effects of its constructs on physical activity intention, results pointed to the existence of positive direct effects of self-identity on intention, as well as on physical activity behavior. What was also identified was the indirect influence of self-identity on behavior, through intentions, which additionally points to the significance of including this variable into the model.

In addition to children, physical activity intention and behavior were researched by use of TPB approach for parents (mothers and fathers) as well. For this purpose, three types of beliefs were used: behavioral, normative and control, corresponding to attitudes, subjective norms and perceived behavioral control, respectively [36]. According to the results of this research, a certain number of the above-mentioned beliefs correlates significantly to parents' physical activity intention and behavior, with several differences from gender aspect. Also, by means of regression analysis, key beliefs and targets, which may feature as a base for creating appropriate strategies of encouraging physical activity among parents were identified.

There can also be identified the application of theory of planned behavior in the context of physical activity of the elderly [37]. Based on a certain number of studies, it was stated that physical activity intention was more explicable by TPB constructs in the elderly in comparison with young adults. Furthermore, subjective norms and perceived behavioral control were singled out as especially significant determinants of intentions and behavior of the elderly, due to which focus on the above-mentioned factors may be useful when promoting exercise to the given target segment.

Theory of planned behavior is frequently used to explain and, to a lesser extent, predict physical activity behavior and healthy nutrition [38] (p. 413). In this context, cited authors devoted attention to perceived behavioral control and self-efficiency, analyzing responses of three segments of users of Michelle Bridges 12 Week Body Transformation—MB12WBT program (depending on the degree of use of the program). Results of their study showed that self-efficiency and perceived behavioral control represent two different constructs. In addition, unlike perceived behavioral control, self-efficiency stands out as a significant predictor of physical activity behavior and healthy nutrition for all three segments, which was not confirmed in the case of the users' intention.

### 2.4. Conceptual Model and Hypotheses Development

Starting from the previously elaborated sources, the model presented in Figure 1 is proposed. Based on the theory of planned behavior, it includes behavioral, normative and control beliefs, which altogether affect physical activity behavioral intention. Moreover, in accordance to the research of Ries et al. [35], the model contains two additional variables, motivation and self-identity.

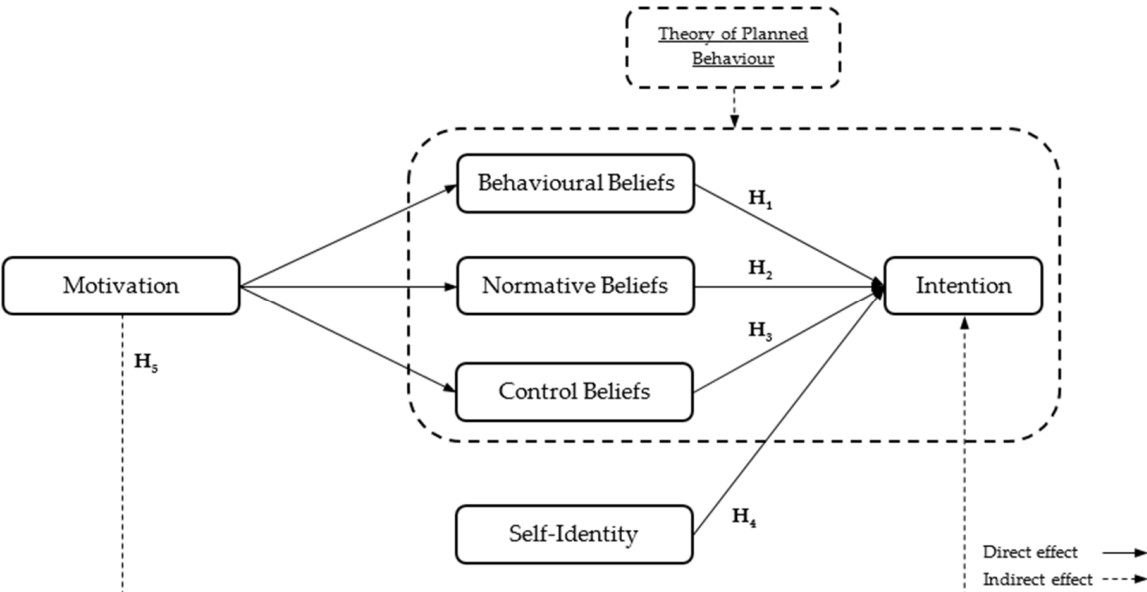

**Figure 1.** Conceptual model—physical activity behavioral intention.

Bearing in mind that outcomes of physical activity include improvements in physical and mental health, loss of weight and better socialization, it is expected that behavioral beliefs have positive influence on behavioral intention. Because engaging in physical activity can be socially supported, behavioral intention can be positively affected by normative beliefs as well. On the other hand, physical activity behavioral intention can be negatively influenced by control beliefs, taking into account certain limitations and barriers, such as cost, lack of time, tiredness, etc. Therefore, when it comes to those three types of beliefs, the following hypotheses were set:

**Hypotheses 1.** *Behavioral beliefs positively and significantly affect behavioral intention.*

**Hypotheses 2.** *Normative beliefs positively and significantly affect behavioral intention.*

**Hypotheses 3.** *Control beliefs negatively and significantly affect behavioral intention.*

As presented in the conceptual model, besides the above-mentioned beliefs, behavioral intention can be directly affected by self-identity. The more prominent this identity (self-perception) is, the more likely that person will consequently behave in accordance to it; in the case of physical activity, it can mean that person who sees herself or himself as sporty or fit is more likely to engage in this type of behavior [35]. Hereby, the fourth hypothesis is:

**Hypotheses 4.** *Self-identity positively and significantly affects behavioral intention.*

In addition to direct effects, the model included indirect effect of motivation on behavioral intention. Motivation was already analyzed in the context of theory of planned behavior as a predictor of its constructs, and by means of them, it influenced intention [35]. Hereby, the following hypothesis was set:

**Hypotheses 5.** *Motivation positively, significantly and indirectly affects behavioral intention.*

## 3. Materials and Methods

The research was conducted on the convenience sample of students from the University of Novi Sad, Serbia. The sample consisted of 231 respondents. They were more than 21 years old on average. There were around 20% of male and 80% of female students (having in mind the willingness to participate in the research and their percentage in total student population). Data were collected in 2019.

The statistical model consists of six constructs: motivation, three types of beliefs (behavioral, normative and control), self-identity and behavioral intention (Figure 2).

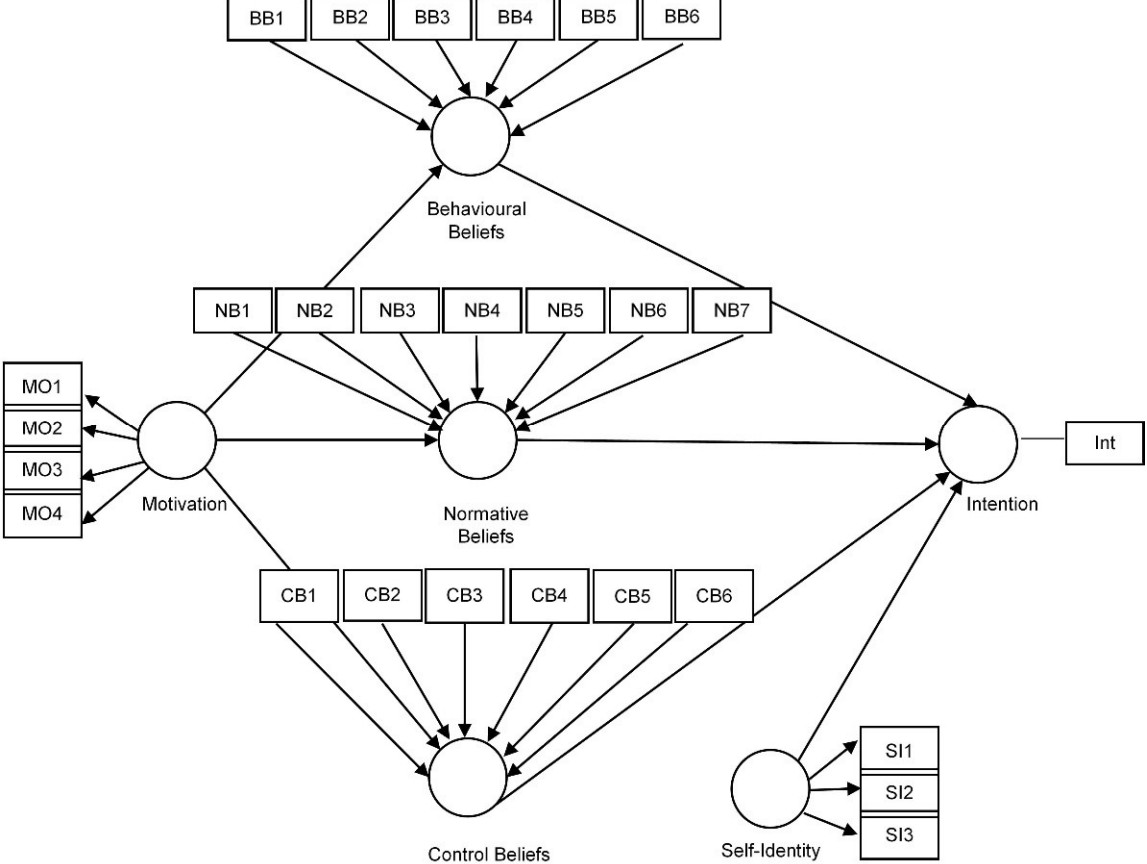

**Figure 2.** Statistical model—physical activity behavioral intention.

The items from the questionnaire can be seen in tables providing the results of its testing (Tables 1 and 3). All the items of all six variables were assessed on five-point Likert scale (from "strongly disagree" to "strongly agree").

Beliefs (behavioral, normative and control) were measured in accordance with previous research [36]. Hereby, items BB3 and BB5 were recoded. In this research, the items measuring beliefs are treated as formative. That is in accordance with contemporary marketing scientific literature [39] and presents methodological contribution of this research. Following recommendations for such modeling [40], in order to assess the properties of the questionnaire related to formative constructs, the authors added three general single items that would reflect each one of the beliefs: "Improve my overall well-being" for behavioral beliefs, "People important to me" for normative beliefs and "Obstacles" for control beliefs.

Motivation and self-identity were measured in accordance with previous researches [35,41]. They were treated as reflective constructs also in accordance to the literature dealing with the differences between formative and reflective constructs [39]. Intention is measured as a single-item construct ("I intend to do regular physical activity in the next week") in accordance to previous research [36].

When analyzing reflective constructs, we have examined internal consistency reliability, individual indicator reliability, convergent validity and discriminant validity [40]. Hereby, for their evaluation, we have used composite reliability (CR), outer loadings, average variance extracted (AVE) and the heterotrait–monotrait ratio (HTMT), respectively. For assessing discriminant validity, we have checked the confidence intervals of HTMT.

When it comes to formative constructs, the assessment procedure includes the analysis of convergent validity, collinearity issues, and relevance and significance of formative

indicators [40]. For testing convergent validity, redundancy analysis was applied. Therefore, three additional structural models have been created. Each model consisted of a formatively measured construct as an exogenous variable and one single-item endogenous variable, which was previously specified. Collinearity of formative indicators was checked by examining their VIF values. After analyzing outer VIF values, significance of each formative indicator's outer weight was tested. In addition, when outer weights of formative indicators are not significant, their outer loadings should be analyzed (their values should be higher or equal to 0.5, or at least significant) [40].

Considering the evaluation of the structural model, it was firstly checked for collinearity issues by analyzing the inner VIF values. After checking the VIF values of predictor constructs, the $R^2$ values of endogenous constructs were examined. Finally, in order to assess the structural model relationships, path coefficients have been analyzed. Path coefficients have been analyzed in the context of gender as well.

## 4. Results

### 4.1. Measurement Model

The first part of the analysis regarding reflective constructs is presented in Table 1.

**Table 1.** Quality criteria of the reflective constructs.

| Constructs and Items | Loadings | AVE | CR |
|---|---|---|---|
| Motivation | | 0.783 | 0.935 |
| MO1—I exercise because it's fun. | 0.773 | | |
| MO2—I enjoy my exercise sessions. | 0.933 | | |
| MO3—I find exercise a pleasurable activity. | 0.936 | | |
| MO4—I get pleasure and satisfaction from participating in exercise. | 0.888 | | |
| Self-identity | | 0.792 | 0.920 |
| SI1—I see myself as sporty. | 0.856 | | |
| SI2—I see myself as fit and healthy. | 0.893 | | |
| SI3—I see myself as physically active person. | 0.919 | | |

As it can be seen, the outer loadings for all indicators are higher than 0.7, the values of AVE (average variance extracted) are higher than 0.5 and the values of CR (composite reliability) are higher than 0.7. The obtained results confirm internal consistency reliability, individual indicator reliability and convergent validity [40]. In addition, AVE and CR values for intention equaled 1 because it was presented as a single-item construct.

The results of testing discriminant validity of reflective constructs are presented in Table 2.

**Table 2.** Discriminant validity HTMT.

| | HTMT | 2.5% | 97.5% |
|---|---|---|---|
| Motivation—Intention | 0.586 | 0.484 | 0.668 |
| Self-Identity—Intention | 0.621 | 0.504 | 0.702 |
| Self-Identity—Motivation | 0.671 | 0.575 | 0.751 |

The lower and upper bounds are presented in columns marked with 2.5% and 97.5%. Bearing in mind that value 1 is outside the all three confidence intervals, the discriminant validity is supported. Moreover, the $HTMT_{.85}$ criterion is satisfied as well, i.e., all presented HTMT values are much below the threshold of 0.85 [42].

When it comes to testing the formative constructs, the results of redundancy analysis are firstly taken into account. Path coefficients in the case of all three models were above the threshold value of 0.70 (0.744, 0.965 and 0.950 for behavioral, normative and control beliefs, respectively), supporting the convergent validity of formative constructs.

In addition, in regards to testing formative constructs, outer VIF values, outer weights and outer loadings are considered (Table 3).

**Table 3.** Outer VIF values, outer weights and outer loadings.

| Constructs and Items | VIF | Outer Weights | Outer Loadings |
|---|---|---|---|
| Behavioral Beliefs | | | |
| BB1—Improve my physical health and fitness. | 1.644 | 0.602 *,1 | 0.661 * |
| BB2—Improve my mental well-being. | 1.592 | 0.345 * | 0.689 * |
| BB3—Increase the risk of sustaining pain/injury. | 1.107 | 0.159 | 0.257 |
| BB4—Give me the opportunity to socialize. | 1.167 | 0.151 | 0.095 |
| BB5—Interfere with my other commitments. | 1.088 | 0.582 * | 0.547 * |
| BB6—Help me to lose weight/control my weight. | 1.228 | −0.168 | 0.058 |
| Normative Beliefs | | | |
| NB1—Partner | 1.520 | 0.054 | 0.397 * |
| NB2—Parents | 1.720 | 0.343 | 0.586 * |
| NB3—Other family members | 2.408 | −0.051 | 0.306 |
| NB4—Friends | 2.302 | −0.087 | 0.345 * |
| NB5—Healthcare professionals | 1.740 | 0.118 | 0.540 * |
| NB6—Colleagues from Faculty | 2.494 | −0.411 | 0.233 |
| NB7—People I exercise with | 1.699 | 0.960 * | 0.891 * |
| Control Beliefs | | | |
| CB1—Lack of time | 1.228 | 0.262 * | 0.526 * |
| CB2—Tiredness and fatigue | 1.341 | 0.282 * | 0.587 * |
| CB3—Inconvenient | 1.334 | 0.411 * | 0.617 * |
| CB4—Lack of motivation | 1.340 | 0.348 * | 0.672 * |
| CB5—Cost | 1.250 | 0.085 | 0.301 * |
| CB6—Illness and injury | 1.105 | −0.520 * | −0.353 * |

[1,]* $p < 0.05$.

As presented in Table 3, all VIF values are lower than the proposed threshold of 5, indicating that there are no collinearity issues. When it comes to significance of each formative indicator's outer weight, it can be seen that for three indicators of behavioral beliefs (BB3, BB4 and BB6), six indicators of normative beliefs (NB1, NB2, NB3, NB4, NB5 and NB6) and one indicator of control beliefs (CB5), p values were higher than 0.05. Therefore, their outer loadings were analyzed. For three indicators of behavioral beliefs (BB3, BB4 and BB6) and two indicators of normative beliefs (NB3 and NB6), outer loadings were both, lower than 0.5 and nonsignificant. Consequently, these indicators have been eliminated from further analysis, and the initial model has been modified.

### 4.2. Structural Model

Having in mind that all inner VIF values were all lower than 5, it could be concluded that there were no issues related to collinearity in this model. In addition, the $R^2$ values of endogenous constructs were examined. They equaled 0.168 for behavioral beliefs, 0.142 for normative beliefs, 0.321 for control beliefs and 0.448 for intention. Path coefficients have been analyzed, and their values are presented in Table 4.

**Table 4.** Path coefficients.

| Path | Direct Effect | Indirect Effect | Total Effect |
|---|---|---|---|
| Motivation → Behavioral Beliefs | 0.410 *,1 | - | 0.410 * |
| Motivation → Normative Beliefs | 0.377 * | - | 0.377 * |
| Motivation → Control Beliefs | −0.566 * | - | −0.566 * |
| Behavioral Beliefs → Intention | 0.272 * | - | 0.272 * |
| Normative Beliefs → Intention | 0.110 * | - | 0.110 * |
| Control Beliefs → Intention | −0.101 | - | −0.101 |
| Self-Identity → Intention | 0.407 * | - | 0.407 * |
| Motivation → Intention | - | 0.210 * | 0.210 * |

[1],* $p < 0.05$.

Path coefficients analyzed in the context of gender are presented in Table 5.

**Table 5.** Path coefficients in the gender context.

| Path | Direct Effect | | Indirect Effect | | Total Effect | |
|---|---|---|---|---|---|---|
| | Male | Female | Male | Female | Male | Female |
| Motivation → Behavioral Beliefs | 0.290 *,1 | 0.437 * | - | - | 0.290 * | 0.437 * |
| Motivation → Normative Beliefs | 0.162 | 0.449 * | - | - | 0.162 | 0.449 * |
| Motivation → Control Beliefs | −0.660 * | −0.564 * | - | - | −0.660 * | −0.564 * |
| Behavioral Beliefs → Intention | −0.009 | 0.306 * | - | - | −0.009 | 0.306 * |
| Normative Beliefs → Intention | 0.140 | 0.123 * | - | - | 0.140 | 0.123 * |
| Control Beliefs → Intention | −0.253 | −0.095 | - | - | −0.253 | −0.095 |
| Self-Identity → Intention | 0.450 * | 0.390 * | - | - | 0.450 * | 0.390 * |
| Motivation → Intention | - | - | 0.187 | 0.243 * | 0.187 | 0.243 * |

[1],* $p < 0.05$.

At the level of the whole sample, motivation has significant positive effects on behavioral beliefs and normative beliefs, while its effect on control beliefs is also significant, though negative. All effects on intention, except for control beliefs, were positive and significant with p lower than 0.05. Hereby, the intention was affected directly by behavioral beliefs, normative beliefs and self-identity, and indirectly by motivation.

When it comes to males, motivation significantly affects behavioral beliefs positively and control beliefs negatively. In addition, none of the beliefs affect intention significantly, while such positive influence exists in the case of self-identity. Finally, there is no significant indirect effect of motivation on intention.

As for females, all beliefs are significantly affected by motivation (behavioral and normative positively and control negatively). Moreover, there is significant and positive indirect effect of motivation on intention. Intention is also significantly positively affected by behavioral and normative beliefs, as well as by self-identity.

## 5. Discussion and Conclusions

### 5.1. Theorethical and Methodological Implications

Generally, the results obtained in this research are in line with previous researches conducted abroad and described within this paper. It should be stressed that authors in this research start from different beliefs (behavioral, normative and control) that correspond to elements of the theory of planned behavior (respectively to attitudes, subjective norms and perceived behavioral control), i.e., direct antecedents of intention to behave, what was also the case in some of the previous researches [34,36]. Out of the listed beliefs, behavioral and normative beliefs affect intention positively and significantly, while control beliefs do not have significant impact on that variable. Therefore, hypotheses $H_1$ and $H_2$ were confirmed, which was not the case with the hypothesis $H_3$. The existence of at least one of the named influences is in accordance to some previous researches [32–37].

Furthermore, it should be noted that the stronger effect on intention to perform physical activities is the one of the self-identity, thus confirming the hypothesis $H_4$. The positive influence of self-identity is also found in other researches [34,35]. In addition, the

hypothesis $H_5$ is confirmed, i.e., motivation influences intention significantly, although indirectly, and has significant influence on all three types of beliefs, the positive in the case of behavioral and normative beliefs, and negative in the case of control beliefs. The inclusion of the variable motivation and the existence of its effects are also present in one previous research [35].

However, having in mind the need for the social marketing to consider the tools of commercial marketing such as market segmentation, it becomes more important to consider the results for genders separately. Such differences are also noticed in some of the previous researches [33,34]. In the concrete case, when it comes to male students, only their self-identity (respondent's perception of himself as sporty, fit and healthy, and physically active person) influences their intention to perform regular physical activity in the week after questioning. Motivation does affect some of their beliefs, but those beliefs do not influence intention, neither is it influenced indirectly by motivation.

When it comes to female students, the explanation of influences on intention to perform regular physical activity in the week after questioning is more complex. The largest positive influence is also by self-identity, i.e., by the level in which the female student sees herself as a sporty, fit and healthy, and physically active person. However, there is also an influence of the similar strength of their behavioral beliefs, i.e., the respondent's perception that if she engages in physical activities, it would likely improve her health and fitness, mental well-being and will not interfere with her commitments. There is also a positive influence of normative beliefs, i.e., respondent's perception that it is likely that her partner, parents, friends, healthcare professionals and people she exercises with, think that she should perform physical activities. Finally, motivation (the choice of the respondents to exercise because it is fun, enjoyable, pleasurable and satisfying) influences behavioral intention indirectly and positively while influencing previously mentioned beliefs directly and positively.

It can be also noticed that control beliefs (the perception of the respondents that it is likely that some of the obstacles (lack of time, tiredness and fatigue, inconvenience, lack of motivation, cost, illness and injury) will prevent them from performing physical activities) are not found as influential on respondents' intention to perform regular physical activity in the week after questioning in neither of the cases. Control beliefs are in each of the cases influenced negatively by motivation; however, because those beliefs have no additional influence, that relation will not be elaborated further.

As for methodological contribution of this research, the authors would stress that, according to their knowledge, this is the first time that in researches with similar topics, behavioral, normative and control beliefs were conceptualized as formative constructs. That was performed in accordance to recommendations for differentiating between formative and reflective constructs [39]. Although not elaborated upon in detail in this paper because it would go out of the scope of the paper, what is often present even in the highly respected marketing literature—especially in the context of model misspecifications leading to totally wrong conclusions—is an important issue [39].

### 5.2. Practical and Managerial Implications

The social marketing implications can be defined at different levels. The authorities, the health professionals, the university as well as the factors from the supply side of physical activity market should consider the obtained results from different perspectives. However, the results should not be considered only from the aspect of promotion but also from the whole marketing mix, especially creating a "customer value" point. In any case, there is no negative influence of perceived obstacles on intention to perform physical activity. Only positive influences should be considered. In the case of male students, the value should be created, distributed and communicated in a manner to support connecting performing physical activity with their self-identity. Because this factor had the largest path coefficient for both genders, special attention should be dedicated to creating a positive image of being a sporty, fit and physically active young person [34]. The possibilities of

targeting in social media era can be of the great help in communicating that message. There is also a possibility to try to encourage male students that are not physically active at the moment to start being active in the future by taking into consideration that result. When it comes to female students, the value for them should also include elements of positive consequences, support from important persons as well as enjoying the exercise process itself.

### 5.3. Future Research Directions

In the future researches, one could try to reach a wider and more representative sample in order to compare students from different universities and countries and to include additional demographic and socio-psychological variables. Moreover, the model could be extended with new variables related to TPB elements in order to gain deeper insights in analyzed relations.

**Author Contributions:** All of the authors formulated goals of the research and interpreted available literature; conducting and analyzing research was performed by N.M. and N.D., while implications were developed by I.D. and A.G. All authors have read and agreed to the published version of the manuscript.

**Funding:** This research received no external funding.

**Institutional Review Board Statement:** Not applicable.

**Informed Consent Statement:** Not applicable.

**Data Availability Statement:** Not applicable.

**Conflicts of Interest:** The authors declare no conflict of interest.

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
