# Peer review of "Social Marketing in Promoting Sustainable Healthy Lifestyle among Student Population"

_sustainability, doi:10.3390/su14031874_

Round 1
Reviewer 1 Report
Dear Author (s),
Although the topic is interesting, the manuscript needs significant revision.
- You need to draw the original conceptual model (not just the statistical one you presented). Also, draw it by yourself so that it will be clearly presentable.
- The hypotheses are presented together at the moment. it should be clearly separated with logical argument.
- H1 & H2 are very complicated. Please make it simple and easily understandable.
- Please add the practical and managerial implications in a separate paragraph with separate headline.
- Add theoretical contribution in a separate paragraph with separate headline.
- Add future research direction in a separate paragraph with separate headline.
Good Luck!
Author Response
Dear reviewer,
We thank you for your comments and have made all changes according to your request.
Best regards,
Authors

Reviewer 2 Report
The paper discusses the encouraging student for physical activity and healthy lifestyle using social marketing. This is an interesting subject to human being. The abstract is needed, some modifications and improvement. In addition, the Introduction is well written as well as the Literature review where the authors presented firstly an explanation of sustainable, healthy lifestyle and the role of physical activity, social marketing and description. The methodology is well explained and clear just the authors must add some sentences from methodology in the abstract (year/time/numbers of respondents..). The results section is clear and support the used methodology. Finally, the discussion and conclusion section is very clear and organized in term of the study objectives.
I have attached here the document with some comments

Author Response

(The authors gave the same response as above.)

Round 2
Reviewer 1 Report
The revised version is ok.